# Targeting PGM3 as a Novel Therapeutic Strategy in *KRAS/LKB1* Co-Mutant Lung Cancer

**DOI:** 10.3390/cells11010176

**Published:** 2022-01-05

**Authors:** Hyunmin Lee, Feng Cai, Neil Kelekar, Nipun K. Velupally, Jiyeon Kim

**Affiliations:** 1Department of Biochemistry and Molecular Genetics, College of Medicine, University of Illinois at Chicago, Chicago, IL 60607, USA; hmlee19@uic.edu (H.L.); nkelek2@illinois.edu (N.K.); nkv2@uic.edu (N.K.V.); 2Children’s Medical Center Research Institute, UT-Southwestern Medical Center, Dallas, TX 75390, USA; feng.cai@utsouthwestern.edu

**Keywords:** cancer metabolism, metabolic vulnerability, non-small cell lung cancer, *KRAS* and *LKB1* co-mutations, the hexosamine biosynthesis pathway, glycosylation, O-GlcNAcylation

## Abstract

In non-small-cell lung cancer (NSCLC), concurrent mutations in the oncogene *KRAS* and tumor suppressor *STK11* (also known as LKB1) confer an aggressive malignant phenotype, an unfavourability towards immunotherapy, and overall poor prognoses in patients. In a previous study, we showed that murine *KRAS/LKB1* co-mutant tumors and human co-mutant cancer cells have an enhanced dependence on glutamine-fructose-6-phosphate transaminase 2 (GFPT2), a rate-limiting enzyme in the hexosamine biosynthesis pathway (HBP), which could be targeted to reduce survival of *KRAS/LKB1* co-mutants. Here, we found that *KRAS/LKB1* co-mutant cells also exhibit an increased dependence on *N*-acetylglucosamine-phosphate mutase 3 (PGM3), an enzyme downstream of GFPT2. Genetic or pharmacologic suppression of PGM3 reduced *KRAS/LKB1* co-mutant tumor growth in both in vitro and in vivo settings. Our results define an additional metabolic vulnerability in *KRAS/LKB1* co-mutant tumors to the HBP and provide a rationale for targeting PGM3 in this aggressive subtype of NSCLC.

## 1. Introduction

Lung cancer is the leading cause of cancer-related mortality worldwide with a 5-year survival rate of only 21% [1]. Of lung cancer cases, NSCLC accounts for about 85% [2]. Through the implementation of large-scale sequencing efforts, tumor suppressors and oncogenes mutated in NSCLC have been defined. However, most of these mutations are not yet subject to targeted therapy. *KRAS* is the most mutated oncogene in lung adenocarcinoma (LUAD), the most prominent subtype of NSCLC [3]. Although there are recently FDA approved small molecules (e.g., sotorasib and adagrasib) [4] acting via covalent modification to target *KRAS* G12C mutants, the majority of KRAS mutations still lack therapeutic targeting. Loss of the tumor suppressor *STK11* encoding the serine/threonine kinase LKB1 occurs in about 25% of NSCLC [5]. Notably, the incidence of concomitant *KRAS* and *LKB1* mutations in NSCLC is estimated to be ~25% of all *KRAS*-driven NSCLC. While patients with these co-mutations have poor clinical outcomes and a more aggressive malignancy that frequently develops brain metastases [6,7], there are currently no therapies available to cure *KRAS/LKB1* (KL) co-mutant NSCLC. Importantly, LKB1 loss drives primary resistance to PD-1 blockade in *KRAS*-driven NSCLC, rendering these tumors unfavorable to immunotherapy [8]. Therefore, there is an unmet need to discover the molecular mechanism by which KL co-mutations alter signal transduction pathways in tumors and determine therapeutic avenues for targeting pathways, specifically without affecting normal tissue physiology.

Metabolic reprogramming is a hallmark of cancer and occurs as tumors alter metabolic pathways to meet the bioenergetic, biosynthetic, and redox requirements of malignancy. The mutation of oncogenes and/or tumor suppressors is a major factor in cell-autonomous metabolic reprogramming. Interestingly, noted liabilities in the KL co-mutant state include an enhanced dependence on the electron transport chain [9], the pyrimidine synthesis enzyme deoxythymidylate kinase [10], lysosomal acidification [11], and the urea cycle enzyme carbamoyl phosphate synthetase 1 (CPS1) [12]. Recently, we found another selective vulnerability of these KL co-mutant cancers to inhibition of glutamine-fructose-6-phosphate transaminase 2 (GFPT2), a rate-limiting enzyme in the amino sugar/nucleotide sugar pathway called the hexosamine biosynthesis pathway (HBP) [13]. In that study, we demonstrated that KL co-mutant cells, compared to *KRAS* mutant cells, show higher HBP activity, KL co-mutant tumor growth, but not *KRAS* mutant tumor growth, is impaired by the GFPT inhibitor azaserine in both subcutaneous xenograft and autochthonous lung tumor mouse models. Although these results provide insight into a therapeutic strategy for KL co-mutant NSCLC, there are no clinically approved drugs to inhibit GFPT2 in a well-tolerated manner. Azaserine inhibits purine biosynthesis and other pathways [14]. Prolonged treatment of azaserine (>3 weeks) induces tumor growth in rats [15,16]. Thus, identifying more effective strategies to target the GFPT2 pathway remains important.

While seeking better therapeutic strategies to target the GFPT2 signaling pathway, we discovered that KL co-mutant cells exhibit an enhanced reliance on *N*-acetylglucosamine-phosphate mutase 3 (PGM3), an enzyme downstream of GFPT2. Genetic suppression and pharmacological inhibition of PGM3 by FR054, a competitive inhibitor of PGM3, impaired tumor growth and promoted the death of KL co-mutant NSCLC cells both in vitro and in vivo. Our study identified PGM3 inhibition as a potentially effective approach to target KL co-mutant NSCLC.

## 2. Materials and Methods

### 2.1. Cell Lines, Culture, and Reagents

All NSCLC cell lines (A549 (*KRAS^G12S^*), H460 (*KRAS^Q61H^*), H2122 (*KRAS^G12C^*), H1373 (*KRAS^G12C^*), and Calu-6 (*KRAS^Q61K^*)) used in this study were obtained from Hamon Cancer Center Collection (University of Texas Southwestern Medical Center). Cells were cultured in RPMI 1640 medium (Sigma-Aldrich, St. Louis, MO, USA, catalog no. R8758), supplemented with 5% fetal bovine serum (Sigma-Aldrich, catalog no. 12306C) and penicillin/streptomycin (Thermo Fisher, Waltham, MA, USA, catalog no. 15140122), at 37 °C in a humidified atmosphere containing 5% CO_2_. All cell lines were verified as mycoplasma free with e-Myco Kit (Bulldog Bio, Portsmouth, NH, USA, catalog no. 2523348). FR054 was obtained from AOBIOUS (catalog no. AOB31146), and azaserine was obtained from Sigma-Aldrich (catalog no. A4142). H460- and H2122-empty vector (EV) and wildtype LKB1 (LKB1) were generated with lentiviral vectors expressing LKB1. LentiV_Neo_LKB1 was from Christopher R. Vakoc (plasmid no. 108111, Addgene). shGFP- and shLKB1-expressing Calu-6 and H1373 cells were generated with lentiviral vectors expressing pLentiX2-PURO-shLKB1-Hu and pLentiX2-shGFP. pLentiX2-PURO-shLKB1-Hu was from Reuben Shaw (plasmid no. 61242, Addgene). To generate CRISPR-Cas9-*PGM3* knockout cell lines, H460, A549, and H2122 cells were infected by LentiCRISPR V2 expressing sgRNA targeting *PGM3* and an empty vector as the control. LentiCRISPR V2 was from Feng Zhang (plasmid no. 52961, Addgene). The following primers were used to generate the sgRNA *PGM3* constructs:

sg*PGM3* no.1

Forward: 5′-CACCGGCGCTTCTCCGCTGCGTGT-3′,

Reverse: 5′-AAACACACGCAGCGGAGAAGCGCC-3′;

sg*PGM3* no.2

Forward: 5′-CACCGACTGCTGGATTTCGAACGA-3′,

Reverse: 5′-AAACTCGTTCGAAATCCAGCAGTC-3′;

sg*PGM3* no.3

Forward: 5′-CACCGATGTTTACAATGCAGCCTAC-3′,

Reverse: 5′-AAACGTAGGCTGCATTGTAAACATC-3′.

### 2.2. Western Blot Analysis

Protein lysates from NSCLC cell lines were prepared in CHAPS buffer (1 mM KCl, 50 mM HEPES (pH 7.4), 0.1% CHAPS), supplemented with phosphatase inhibitor (Roche, Basel, Switzerland, catalog no. 4906837001) and protease inhibitor cocktail (Thermo Fisher, catalog no. A32955). Protein lysates were quantified with the BCA protein assay kit (Thermo Fisher, catalog no. 23227). Samples were separated by 7% SDS-PAGE gels, transferred to PVDF membranes, and probed with antibodies against PGM3 (1:1000, Santa Cruz, catalog no. SC-100410), LKB1 (1:1000, Cell Signaling, Danvers, MA, USA, catalog no. 3050), Vinculin (1:2000, Sigma-Aldrich, catalog no. V9131), and β-actin (1:5000, Sigma-Aldrich, catalog no. A1978). Target bands were detected with the ECL western blotting system (Thermo Fisher, catalog no. 32106).

### 2.3. Lectin Binding Analysis

Lectin binding was analyzed by flow cytometry. The following reagents were used for this study: 5 μg/mL of biotin-conjugated LEA (Vector Laboratories, Burlingame, CA, USA, catalog no. B-1175), 10 μg/mL of rhodamine conjugated L-PHA (Vector Laboratories, catalog no. RL-1112), 5 μg/mL of biotin-conjugated SNA (Vector Laboratories, catalog no. B-1305), and 5 μg/mL of streptavidin-conjugated allophycocyanin (SA-APC, Thermo Fisher Scientific, catalog no. S-868). Cells were collected by centrifugation (3 min at 300× *g*), washed twice with Dulbecco’s PBS (DPBS, Sigma-Aldrich, catalog no. D8537), then resuspended in DPBS at 2.0 × 10^6^ cells/mL. After 100 μL of cell suspension, cells were added into a V-shape bottom 96-well plate and spun down at 600 *g* for 5 min. Cells were incubated with 100 μL of lectin, diluted in DPBS for 30 min at 4 °C (L-PHA incubated for 60 min), then washed three times with DPBS. Cells were then incubated with 5 μg/mL of SA-APC in DPBS for 30 min at 4 °C. Cells were analyzed by flow cytometry (CytoFLEX, Beckman coulter) with dual lasers at 488 and 635 nm. Mean fluorescence intensity (MFI) was analyzed by Flow Jo V10.

### 2.4. Wheat Germ Agglutinin (WGA) Pull down Assay

Cells were lysed in CHAPS buffer, and cell lysates were collected by centrifugation. The protein concentrations were quantified with the BCA Protein Assay Kit. To remove sugar from WGA, WGA-conjugated agarose beads were incubated with 0.5 M *N*-Acetyl-D-Glucosamine (pH 3.0) for 30 min at 4 °C. In total, 0.5 mg of lysates per sample was then incubated with 50 μL of WGA for 16 h at 4 °C, and the bound proteins were eluted from the WGA by incubating with 0.5 M *N*-Acetyl-D-Glucosamine (pH 3.0) for 30 min at 4 °C. The eluted proteins were separated by 8% SDS-PAGE gels, transferred to PVDF membranes. The PVDF membrane was washed twice with PBS containing 2% TWEEN-20 for 2 min at room temperature. PVDF was incubated with 5 μg of peroxidase conjugated WGA (Sigma-Aldrich, catalog no. L3892) in 5 mL of PBS containing 1 mM CaCl_2_, 1 mM MnCl_2_, and 1 mM MgCl_2_ for 16 h at room temperature. Target bands were detected with the ECL western blotting system.

### 2.5. Cell Proliferation and Death

To measure cell proliferation, 3 × 10^3^ cells were seeded in a 96 well plate per well in 200 μL RPMI base medium under the conditions described in each experiment. After 3 (azaserine and FR054 treatment) or 4 days (silencing experiments), 30 μL of CellTiter Glo reagent (Promega, Madison, WI, USA, catalog no. G7573) was added to each well and mixed for 30 min at room temperature, and the luminescence was read on a luminometer (FLUOstar Omega, BMG labtech). To test cell death, cells were treated as described in the figure legends, stained with 0.5 μg/mL of propidium iodide (PI) (Thermo Fisher, catalog no. P21493) and 2.5 μL of fluorescein isothiocyanate-conjugated Annexin V (BD Biosciences, catalog no. 556419) and analyzed by flow cytometry (CytoFLEX, Beckman Coulter). Cell death was detected in xenografts using the In Situ Cell Death Detection kit, Fluorescein (Sigma-Aldrich), according to the manufacturer’s protocol. In brief, tissue sections were deparaffinized with xylene and rehydrated with ethanol. After washing in PBS, sections were incubated with reaction solution for 1 h at 37 °C in a humidified atmosphere in the dark. Images were acquired with a Zeiss LSM 700 microscope and quantified by Matlab. In brief, R-G-B values of each image were examined, then two independent filters based on R-G-B values, one highlighting blue (DAPI) stained cells and the other highlighting green and blue (TUNEL) stained cells, were created. The total number of blue (DAPI) and blue-and-green (TUNEL) stained cells for each image was calculated using these two independent filters.

### 2.6. RNA Interference

Gene silencing was performed with endoribonuclease-prepared siRNAs (esiRNA, Sigma-Aldrich) targeting *GFPT2* (catalog no. EHU144601) and *PGM3* (catalog no. EHU030501). Cell viability assays were performed 96 h after single transfection, whereas cell death analysis, ^15^N-glutamine labeling, and all Western blots were performed 48 h after double transfection (two consecutive days of forward transfection).

### 2.7. Soft-Agar Colony-Formation Assay

*PGM3* knockout cells (1000 and 1500 per well in a 12 well plate) or FR054 treated cells (2000 and 5000 per well in a 6 well plate) were suspended in 0.375% agar (BD Difco, catalog no. 214220), and pre-equilibrated with growth medium over a 0.75% bottom agar layer in each well. Colonies were allowed to form for 21 days (A549, H460, H2122) or 30 days (H1373), with medium supplementation (a few drops every 5 days). Images were detected with Celigo (Nexcelom). Colony numbers were quantified using Clickmaster2000 v.1.0.

### 2.8. Mouse Xenografts and Tumor Tissue Analysis

Animal procedures were performed with the approval of the University of Illinois at Chicago (UIC) Institutional Animal Care and Use Committees. Mice were euthanized before reaching one of the following humane end points in accordance with UIC guidelines: (1) tumor size greater than 2 cm in diameter; (2) tumor size exceeding 15% of the preoperative body weight as estimated by determining volume (mm^3^) = (length (mm) × width (mm)^2^)/2 and assuming a tumor density of 1 g cm^−^^3^, therefore assuming 1 mm^3^ = 1 mg; (3) >20% net body weight loss; (4) tumor ulceration. 1 × 10^6^ of H1373-shGFP and shLKB1 cells were suspended in serum free RPMI and mixed with Matrigel (Corning, Corning, NY, USA, catalog no. 356237) at 1:1 ratio. Cells were implanted subcutaneously into 6-week-old Athymic Nude-Foxn1nu (Envigo). After tumor cell injection, mice were randomized and then divided into cages. For FR054 treatment, tumor-bearing mice were intraperitoneally injected with FR054 at 500 mg/kg/b.i.d or a vehicle for 14 days after mice presented detectable tumors. Tumor size was measured every 3 days with electronic calipers. When mice were euthanized, whole tumor tissues were collected, and half of them were formalin fixed/paraffin embedded (FFPE). The FFPE tumor sections (5 μm thickness) were then stained with Ki67 antibody (Proteintech, catalog no. 27309-1-AP) at the UIC Research Histology and Tissue Imaging Core. Images were acquired with a Leica DMi8 microscope, and Ki67+ cells were quantified with Matlab. Briefly, images were converted from R-G-B to Hue-Saturation-Value (purity-vibrancy-brightness, H-S-V), and the H-S-V values were examined. Two independent filters based on H-S-V values, one filter highlighting only H&E-stained cells (blue) and the other filter highlighting only Ki67+ cells (brown and blue), were then created, and the total number of blue and brown stained cells for each image was calculated.

### 2.9. ^15^N-Glutamine Labeling

1 × 10^6^ of cells were plated in 6 cm dishes for 16 h before labeling. Cells were incubated in RPMI 1640 media, supplemented with 5% dialyzed fetal bovine serum (Gemini Bio-Products, West Sacramento, CA, USA, catalog no. 100–108) containing 2 mM [γ-^15^N]glutamine (Cambridge Isotope Laboratories, Tewksbury, MA, USA, catalog no. NLM-557) for the indicated times. For the intracellular metabolite analysis by LC-MS/MS, cells were washed with ice-cold saline and then scraped in the presence of 80% acetonitrile (Sigma Aldrich, catalog no. AX0145): 20% LC-MS grade water (Sigma Aldrich, catalog no. WX0001). The cell lysates were subjected to three freeze-thaw cycles between liquid nitrogen and 37 °C and then vortexed for 2 min. The lysates were centrifuged for 15 min at 4 °C, and the supernatants were transferred to a new tube and the centrifugation was repeated. The supernatants (100 µL) were transferred into LC-MS vials. Samples were randomized and analyzed with LC–MS/MS in a blinded manner. LC–MS/MS was performed on an Applied Biosystems 5500 QTRAP liquid chromatography/mass spectrometer equipped with a vacuum degasser, a quaternary pump, an autosampler, a thermostat-equipped column compartment, and a triple quadrupole/ion trap mass spectrometer with electrospray ionization interface, controlled by the Applied Biosystem Analyst software v.1.6.1. A SeQuant ZIC-pHILIC (150 × 2.1 mm^2^, 5 μm) PEEK-coated column was used. Solvents for the mobile phase were 10 mM of NH_4_Ac in H_2_O (pH 9.8, adjusted with concentrated NH_4_OH) (A) and acetonitrile (B). The gradient elution was: 0–15 min, linear gradient 90–30% B; 15–18 min, linear gradient 30% B, 18–19 min, linear gradient 30–90% B; and finally reconditioning it for 9 min using 90% B; The flow rate was 0.25 mL/min and the injection volume was 20 μL. Columns were operated at 40 °C. Declustering potential (DP), collision energy (CE), and collision cell exit potential (CXP) were optimized for each metabolite by direct infusion of reference standards using a syringe pump before sample analysis. Multiple reaction monitoring data were acquired with the following transitions: *N*-acetyl hexosamine-1/6-phosphate: 300/79 (CE, −80 V); uridine diphosphate *N*-acetyl hexosamine: 608/204 (DP, 80 V; CE, 12 V) and 282/79 (DP, −280 V, in-source fragmentation; CE, −72 V). UDP-GlcNAc was not separated from UDP-GalNAc in our liquid chromatography-mass spectrometry (LC–MS) system, although the UDP-HexNAc peak was almost entirely composed of UDP-GlcNAc; Neu5NAc: 310/292 (CE, 8 V). Chromatogram review and peak area integration were performed using the MultiQuant software v.2.1. The peak area for each metabolite was normalized against the total cell number. The normalized area values were used for statistical analysis.

### 2.10. Metabolomics

Cells were incubated in fresh RPMI 1640 media for 2 h, washed with ice-cold saline, and then scraped in the presence of 80% acetonitrile. The cell lysates were subjected to three freeze-thaw cycles between liquid nitrogen and 37 °C and then vortexed for 2 min. The lysates were centrifuged for 15 min at 4 °C, and the supernatants were transferred to a new tube and the centrifugation was repeated. The supernatants (100 µL) were transferred into LC-MS vials. Samples were randomized and analyzed with LC–MS/MS in a blinded manner. LC–MS/MS was performed as described in the ‘^15^N-glutamine labeling’ section.

### 2.11. Patient Survival Data

For prognosis analysis in bladder and breast tumors, *PGM3* mRNA expression data were obtained from The Cancer Genome Atlas and analyzed with OncoLnc (http://www.oncolnc.org, accessed on 21 November 2021). Methods for data generation, normalization, and bioinformatics analyses were previously described in the publication [13]. For the NSCLC, data from this cohort was analyzed using Lung Cancer Explorer (http://lce.biohpc.swmed.edu, 21 November 2021).

### 2.12. Statistics

No statistical methods were used to predetermine sample size. For xenograft experiments, mice injected with tumor cells were randomized before being allocated to cages. GraphPad PRISM 9 software was used for statistical analysis. Error bars, *p* value, and statistical tests are described in figure legends. A comparison of two mean values was evaluated by the two-tailed unpaired *t*-test. A comparison of multiple mean values was evaluated by one-way ANOVA with Tukey’s multiple comparisons test. To examine the significance in xenograft tumor growth, two-way ANOVA was followed by Tukey’s multiple comparisons test.

## 3. Results

### 3.1. GFPT2 Inhibition Selectively Reduces Glycosylation of KL Co-Mutant Cells

Our previous study demonstrated that inhibition of GFPT2 selectively impaired the viability of KL co-mutant cells, but not *KRAS* mutant/*LKB1* wildtype (K mutant) cells [13]. UDP-*N*-acetylglucosamine (UDP-GlcNAc), a final product in the HBP, is required for O-linked and N-linked glycosylation of membrane proteins, especially for both initiation of *N*-glycosylation in the endoplasmic reticulum (ER) and *N*-glycan branching and antenna elongation in the Golgi apparatus [17,18] (Figure 1A). GlcNAc-branched *N*-glycans are recognized by the lectin phytohemagglutinin-L (L-PHA) [19], while the poly *N*-acetyllactosamine (polyLacNAc) extension of *N*-linked glycan antennae is recognized by the lectin *Lycopersicon esculentum* (LEA). Sialic acid (Neu5Ac, a metabolite further downstream of the HBP, Figure 1A), linked to either *N*-acetylgalactosamine (GalNAc) or galactose, is recognized by the lectin *Sambucus nigra* (SNA) (Figure 1C). The aforementioned glycan structures have been shown to be more abundant in aggressive tumor types [20,21,22,23]. Thus, we hypothesized that such selective growth defects of KL co-mutant cells caused by GFPT2 inhibition are associated with a selective reduction of the tumor-associated glycans in KL cells. To test this hypothesis, we used two isogenic pairs of KL co-mutant cells (H460, H2122) expressing either wildtype LKB1 or an empty vector (EV) control (Figure 1B) and measured surface binding of L-PHA in these cells by FACS analysis. Both *GFPT2* silencing and pharmacological GFPT inhibition (azaserine) selectively reduced tumor-associated glycan structures in KL co-mutant cells (Figure 1D,E). Further, silencing *LKB1* in K mutant cells also led to an increase in *N*-glycan structures (Figure 1F, LKB1 expression is shown in Figure 2B). Collectively, these data demonstrate that GFPT2 suppression selectively impacts glycosylation in KL co-mutant cells, which is consistent with the selective vulnerability of KL cells to GFPT2 inhibition.

### 3.2. PGM3 Inhibition Also Selectively Reduces Glycosylation of KL Co-Mutant Cells

While we previously provided insight into targeting GFPT2 as a therapeutic strategy for KL co-mutant NSCLC [13], there are no clinically approved drugs to inhibit GFPT2 in a well-tolerated manner. UDP-GlcNAc is critical for both glycosylation of membrane proteins and O-GlcNAcylation of intracellular proteins. Based on our results showing the link between GFPT2 inhibition-mediated growth defects and reduction of glycosylation [13] (Figure 1), we hypothesized that inhibiting these glyco-functionalization pathways may mimic GFPT2 inhibition (e.g., azaserine treatment).

PGM3 catalyzes the conversion of *N*-acetylglucosamine-6-phosphate (GlcNAc-6-*p*) into *N*-acetyl glucosamine-1-phosphate (GlcNAc-1-P) (Figure 2A). Its modulation controls both O-GlcNAcylation and glycosylation, and reduction of PGM3 activity shows a decrease in the HBP flux leading to suppression of complex *N*-glycans and O-GlcNAcylation levels [24,25]. First, to examine the effect of *PGM3* silencing on the HBP, we performed targeted metabolomics to detect the abundance of the HBP metabolic intermediates. *PGM3* silencing increased intracellular levels of GlcNAc-6-P, the metabolite upstream of PGM3, but decreased those of two PGM3 downstream metabolites, UDP-HexNAc (a surrogate of UDP-GlcNAc), and Neu5Ac (Figure 2C, Appendix A), confirming the functional impact of *PGM3* knockdown. Next, to test whether PGM3 suppression impacts glycosylation in KL co-mutant cells more effectively than in K mutant cells, we used three isogenic pairs of NSCLC cells (two KL co-mutant cell lines (H460, H2122) and one K cell line (H1373)), silenced *PGM3* (Figure 2B), and probed the overall number of glycosylated proteins by using wheat germ agglutinin (WGA), a lectin that binds GlcNAc moieties. Compared with LKB1-expressing cells (the K mutant mimicking condition), *PGM3* silencing in H460 cells lowered the number of glycosylated proteins (Figure 2D). *PGM3* silencing also significantly decreased the number of tumor-associated glycan structures in KL co-mutant but not K mutant cells (Figure 2E,F). These data indicate that the genetic suppression of *PGM3* perturbs intracellular levels of the HBP metabolic intermediates in KL co-mutant cells, leading to the reduction of both O-GlcNAcylation and the abundance of tumor-associated glycan structures.

### 3.3. LKB1 Loss Confers PGM3 Dependence in KRAS-Mutant Lung Cancer Cells

We previously showed that LKB1 status determines cellular sensitivity to GFPT2 suppression; KL co-mutant cells become resistant to GFPT2 suppression when *LKB1* is introduced, whereas K mutant cells exhibit GFPT2 dependence when *LKB1* is silenced [13]. To test whether PGM3, similar to GFPT2, is selectively important for KL co-mutant cell survival, we silenced *PGM3* in three isogenic pairs of NSCLC cells and assessed the effect of PGM3 suppression on cell viability. By comparison to K mutant cells, *PGM3* silencing more potently suppressed the growth of KL co-mutant cells (Figure 3A–C). Such growth defects by *PGM3* silencing were related to increased cell death. As shown in Figure 3D–G, *PGM3* silencing markedly enhanced apoptotic cell death in KL co-mutant cells, but less so in K mutant cells. To examine the role of PGM3 in anchorage-independent cell growth, indicative of the clonogenicity of cancer cells, *PGM3* was deleted by CRISPR-mediated genome editing (CRISPR-V2 lentiviral system: pools, not clones, of KO cells were generated). In all three KL co-mutant cells tested, deletion of *PGM3* significantly suppressed anchorage-independent cell growth (Figure 3H,I).

### 3.4. PGM3 Inhibitor FR054 Reduces the HBP Flow in Both KL and K Cells but Selectively Suppresses Viability of KL co-Mutant Cells

Recent studies reported that FR054, a specific inhibitor of PGM3, has an anti-cancer effect in breast and pancreatic cancer models, both in vitro and in vivo without having severe adverse effects [26,27,28]. Before examining the anti-cancer effect of FR054 in NSCLC, we aimed to confirm whether FR054 effectively inhibits PGM3 activity. The amide (γ) nitrogen donated by glutamine through the GFPT reaction is transmitted to downstream amino sugar metabolites of the HBP, including UDP-HexNAc and *N*-acetylmannosamine (ManNAc) (Figure 4A). Thus, to evaluate the effect of FR054 on the HBP flow, we treated two isogenic pairs (H460, H1373) with FR054 and measured the transfer of the ^15^N from [amide-^15^N]glutamine ([γ-^15^N]glutamine) to UDP-HexNAc and ManNAc (2 h and 6 h time points of isotope labeling for H460 and 6 h time point for H1373). Consistent with our previous study [13], KL co-mutant cells showed higher ^15^N enrichment to the HBP metabolites compared to K mutant cells, implying enhanced HBP activity in KL co-mutant cells. Importantly, while FR054 significantly reduced the HBP flow in both K and KL cells, the abundance of UDP-HexNAc was reduced only in KL co-mutant but not K mutant cells, suggesting that KL co-mutant cells depend more on hexosamine biosynthesis for maintaining the HBP metabolite pools compared to K mutant cells (Figure 4B,C and Appendix A). Consistent with the decreased levels of HBP metabolites in KL co-mutant cells, FR054 reduced O-GlcNAcylation (Figure 4D) and tumor-associated glycan structures (Figure 4E–K) only in KL co-mutant cells and not in K mutant cells.

These results led us to hypothesize that FR054′s selective suppression of glyco-functionalization pathways in KL co-mutant cells may result in a selective sensitivity of KL cells to FR054 treatment. To test the hypothesis, four isogenic pairs of human NSCLC cells and one murine cancer cell pair, with or without LKB1 expression, were treated with FR054. FR054 treatment more potently suppressed growth and increased apoptosis in KL co-mutant cells than in K mutant cells (Figure 5A–G), similar to our *PGM3* silencing results, which show that KL co-mutant cells are more dependent on PGM3 for survival (Figure 3). Likewise, FR054 treatment decreased colony formation of KL co-mutant cells more significantly than K mutant cells (Figure 5H). Our *PGM3* genetic suppression and pharmacological inhibitor data collectively show that KL co-mutant cells require PGM3 for survival to a greater degree than K mutant cells.

### 3.5. PGM3 Inhibition Reduces KL Co-Mutant Tumor Growth In Vivo

To examine the effect of PGM3 inhibition on tumor growth in vivo, nude mice were subcutaneously injected with an isogenic pair of H1373 with or without LKB1 expression. Mice were treated with FR054 (500 mg/kg/dose, twice a day) when palpable tumors were detected (70~100 mm^3^) (Figure 6A). Consistent with previous studies [29], KL co-mutant cells generated bigger tumors compared to K mutant cells. FR054 treatment rarely affected mouse body weight (Figure 6C). Notably, it attenuated the growth of KL co-mutant tumors to a level similar to K tumors, whereas the same treatment enhanced the growth of K mutant tumors (Figure 6B). Reduced tumor burdens were related to both decreased proliferation and increased apoptosis assessed by Ki67 staining and TUNEL assay, respectively (Figure 6D–G). Finally, we examined whether *PGM3* expression is correlated with patient survival. High levels of *PGM3* mRNA correlate with poor prognosis in not only NSCLC and breast cancer [26] but also bladder cancer in a human cohort (Figure 6I,J), suggesting that FR054 treatment may have therapeutic potential for a broader range of tumor types.

## 4. Discussion

NSCLC with concurrent mutations in the oncogene *KRAS* and the tumor suppressor *LKB1* is refractory to most therapies and has the worst predicted outcomes. In our previous study, we showed that KL co-mutant tumors require GFPT2 in the HBP for their survival and proliferation. However, the lack of an effective and safe GFPT2 inhibitor is a major obstacle to the translation of our research findings into clinical practice. In the present study, we provide evidence that PGM3 suppression phenocopies GFPT2 suppression and has therapeutic potential for KL co-mutant tumors (Figure 6H). While we demonstrated that LKB1 loss in the context of oncogenic *KRAS* mutations creates PGM3 dependence, the molecular mechanism by which LKB1 regulates PGM3 needs to be elucidated. In the previous study, we postulated that AMPK may be involved in the regulation of the HBP dependence, as we noted that constitutive AMPK activity overcomes the GFPT2 dependence [13]. Given that GFPT2 and PGM3 are in the same pathway, LKB1 may also regulate PGM3 dependence via AMPK. However, should AMPK be involved in PGM3 regulation, it is less likely that AMPK controls PGM3 by direct phosphorylation; no well-established AMPK consensus motif was found on PGM3 (data not shown). Another possibility would be that other LKB1 downstream kinases participate in PGM3 regulation. The AMPK-related kinases, salt-induced kinase (SIK) 1 and 3, were recently reported to mediate key tumor suppressive effects of LKB1 in NSCLC [30,31]. Further, it has been reported that LKB1 loss-mediated epithelial-to-mesenchymal transition (EMT) occurs through the downstream kinases MARK1 and 4 [32], not AMPK, suggesting an AMPK-independent signaling pathway that is essential for LKB1-dependent control of tumor aggressiveness. Regardless, it will be worthwhile to investigate which downstream kinase(s) is involved in PGM3 regulation and how PGM3 activity is regulated by the LKB1 downstream kinase(s).

It has been reported that PGM3 suppression in breast cancer cells leads to cell death through the induction of endoplasmic reticulum (ER) stress and reactive oxygen species (ROS) [26]. However, in the present NSCLC study, neither PGM3 silencing nor FR054 treatment induced ER stress (Data not shown), suggesting that the underlying death mechanisms may be different, possibly due to tumor heterogeneity derived by different tissue of origin. Although it is beyond the scope of this study, it would be interesting to interrogate the molecular mechanism by which PGM3 inhibition leads to cell death in KL co-mutant NSCLC.

While we showed tumor reduction by FR054 monotherapy, investigating combination drug treatments would allow us to determine whether targeting the pathway enhances the efficacy of cytotoxic chemotherapeutic drug(s). It has been reported that transcriptional regulators (e.g., PRC1 and 2) and histones are O-GlcNAcylated [33]. Should FR054 be proven to perturb chromatin accessibility (e.g., histone H3K27 acetylation) by ATACseq and H3K27Ac-ChIP seq, it would be worth exploring whether FR054 treatment can improve the therapeutic efficacy of FDA-approved epigenetic drugs, such as 5-azacitidine or romidepsin. Collectively, our findings elucidate another metabolic liability in KL co-mutant lung cancer, PGM3 dependence, and explore how PGM3 inhibition (e.g., FR054) can be used to exploit this vulnerability.

## Figures and Tables

**Figure 1 cells-11-00176-f001:**
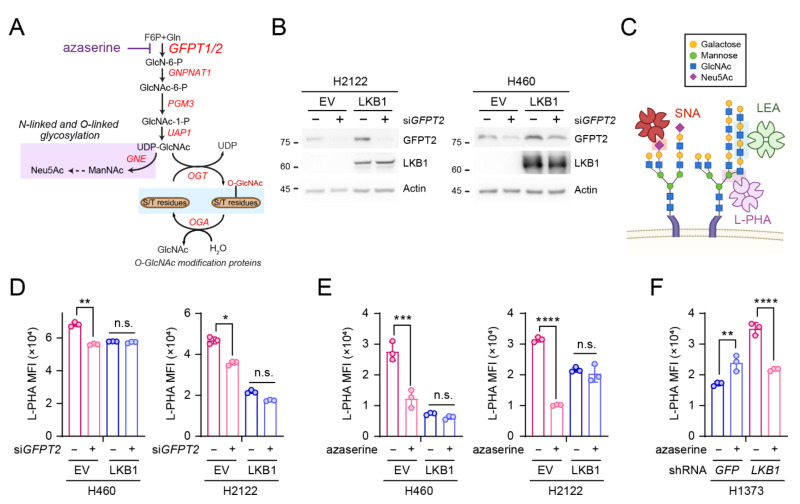
GFPT2 inhibition selectively reduces glycosylation of KL co-mutant NSCLC cells. (**A**) Schematic of the HBP, including azaserine, the GFPT inhibitor. Metabolites are in black and enzymes are in red. Metabolites in the glycosylation pathway are shaded in lilac and those in O-GlcNAcylation are shaded in light blue. F6P, fructose-6-phosphate; Gln, glutamine. (**B**) GFPT2 and LKB1 expression levels were measured in H2122 (**left**) and H460 (**right**) KL co-mutant cells depleted of GFPT2 using endoribonuclease-prepared siRNA (esi*GFPT2*). Actin was used as the loading control. (**C**) Schematic for cell-surface *Sambucus nigra* (SNA), *Lycopersicon esculentum* (LEA), phytohemagglutinin-L (L-PHA) lectins, and corresponding glycan structures, where each lectin interacts. Symbol nomenclature for glycans is shown. (**D**) Cell surface L-PHA lectin binding was measured by flow cytometry in empty vector (EV)- and LKB1-expressing H460 (**left**) and H2122 (**right**) KL co-mutant cells depleted of GFPT2 by esi*GFPT2*. (**E**) Cell surface L-PHA lectin binding was measured by flow cytometry in EV- and LKB1-expressing H460 (**left**) and H2122 (**right**) KL co-mutant cells treated with azaserine (1 µM, three days). (**F**) Cell-surface L-PHA lectin binding was measured by flow cytometry in sh*GFP*- (control) and sh*LKB1*-expressing H1373 K mutant cells treated with azaserine (1 µM, three days). Mean fluorescence intensity (MFI). (**D**–**F**) Statistical significance was assessed using two-tailed Student’s *t*-test/each isogenic pair. n.s., not significant; * *p* < 0.05; ** *p* < 0.01; *** *p* < 0.001; **** *p* < 0.0001. FACS analyses were performed twice, and western blots were repeated three or more times.

**Figure 2 cells-11-00176-f002:**
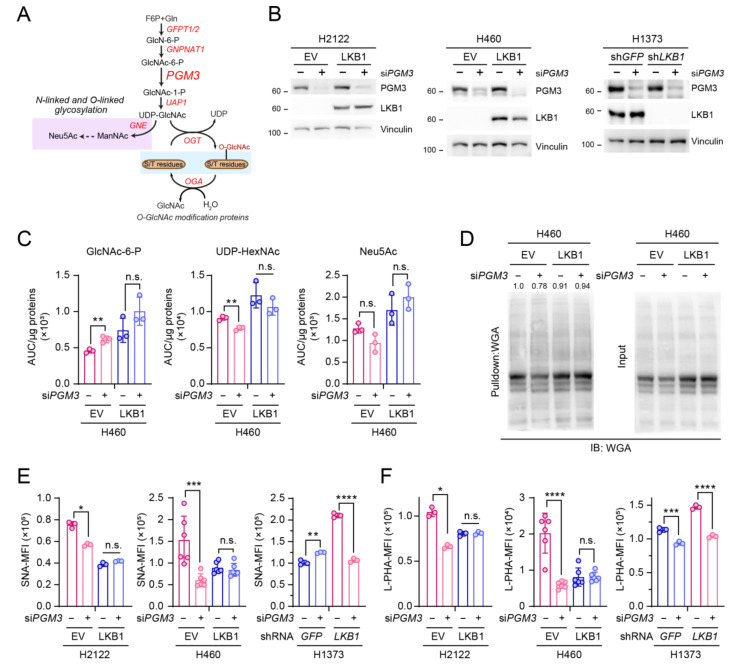
PGM3 inhibition also selectively reduces glycosylation of KL co-mutant NSCLC cells. (**A**) Schematic of the HBP. Metabolites are in black and enzymes are in red. Metabolites in the glycosylation pathway are shaded in lilac, and those in O-GlcNAcylation are shaded in light blue. (**B**) PGM3 and LKB1 expression levels were measured in EV- and LKB1-expressing H2122 (**left**) and H460 (**middle**) KL co-mutant cells and in sh*GFP*- and sh*LKB1*-expressing H1373 (**right**) K mutant cells, depleted of PGM3 using esiRNA targeting *PGM3*. Vinculin was used as the loading control. (**C**) Abundance of hexosamine metabolites in an isogenic pair of H460 KL co-mutant cells depleted of PGM3 by esi*PGM3*. Area under curve (AUC). (**D**) Wheat germ agglutinin (WGA) coupled with agarose was used to precipitate glycosylated proteins from EV- and LKB1-expressing H460 (**left**) KL co-mutant cells depleted of PGM3 by esi*PGM3*. Precipitated proteins were subsequently separated by SDS-PAGE then imaged. Band intensity was quantified with Photoshop, and relative band intensity was obtained by calculating a ratio between each WGA pulldown and input control. Total protein extract before the addition of WGA was used as input control. (**E**,**F**) Cell-surface SNA (**E**) and L-PHA (**F**) lectin binding was measured by flow cytometry in EV- and LKB1-expressing H2122 (**left**) and H460 (**middle**) and in sh*GFP*- and sh*LKB1*-expressing H1373 (**right**) cells depleted of PGM3 by esi*PGM3*. Mean fluorescence intensity (MFI). (**C**,**E**,**F**) Statistical significance was assessed using two-tailed Student’s *t*-test/each isogenic pair. n.s., not significant; * *p* < 0.05; ** *p* < 0.01; *** *p* < 0.001; **** *p* < 0.0001. FACS analyses were performed three times. Targeted metabolomics and WGA pulldown assays were performed once.

**Figure 3 cells-11-00176-f003:**
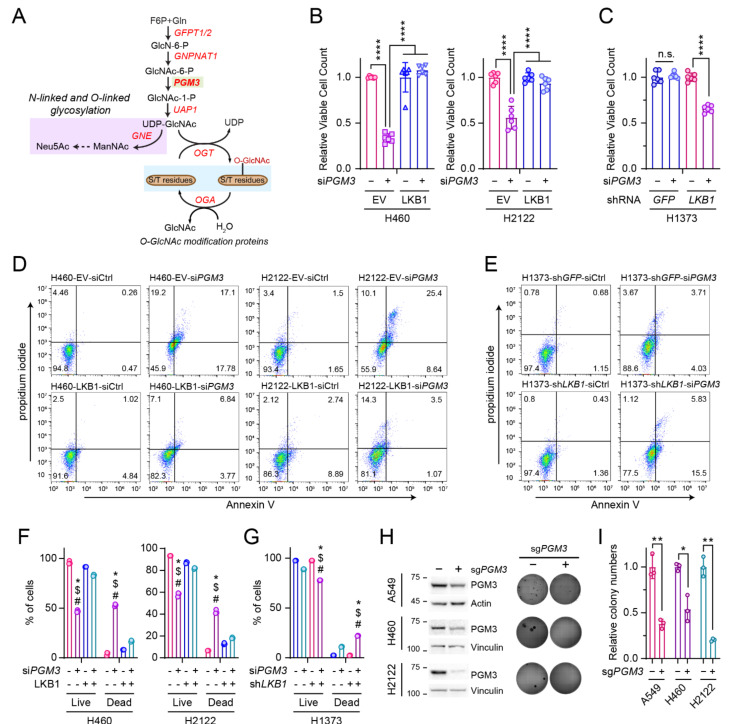
KL co-mutant NSCLC cells require PGM3 for survival. (**A**) Schematic of the HBP. Metabolites are in black and enzymes are in red. Metabolites in the glycosylation pathway are shaded in lilac and those in O-GlcNAcylation are shaded in light blue. (**B**,**C**) Sensitivity to *PGM3* silencing in K mutant and KL co-mutant cells. Two isogenic pairs of KL co-mutant cells (**B**) and one isogenic pair of K mutant cells (**C**) were used. (**D**–**G**) Effect of *PGM3* silencing on cell death in NSCLC cells. (**D**,**E**) Representative dot plots of Annexin V/PI-stained cells with or without *PGM3* silencing. (**F**,**G**) Quantified data from triplicates/cell line tested in (**D**,**E**,**H**); Left*:* abundance of PGM3 in parental and *PGM3* knockout cells. Three KL co-mutant cells were used. Right*:* Effect of *PGM3* knockout on anchorage-independent growth of KL co-mutant cells. Representative images of colony formation assay (*n* = 3). (**I**) Quantified data from (**H**). (**B**,**C**) Statistical significance was assessed using two-tailed Student’s *t*-test/each isogenic pair. **** *p* < 0.0001. (**F**,**G**) Statistical significance was assessed using one-way ANOVA with Tukey’s multiple comparisons test. ^#^ *p* < 0.0001, compared to EV, with control siRNA transfection (**F**) or sh*GFP* with control siRNA transfection (**G**). ^$^ *p* < 0.0001, compared to LKB1 with control siRNA transfection (**F**) or sh*LKB1* with control siRNA transfection (**G**). * *p* < 0.0001, compared to LKB1, with *PGM3* siRNA transfection (**F**) or sh*LKB1* with *PGM3* siRNA transfection (**G**). (**I**) Statistical significance was assessed using two-tailed Student’s *t*-test/each pair of cell line. * *p* < 0.05; ** *p* < 0.01. Western blots, FACS analyses, and soft agar assay were all performed twice.

**Figure 4 cells-11-00176-f004:**
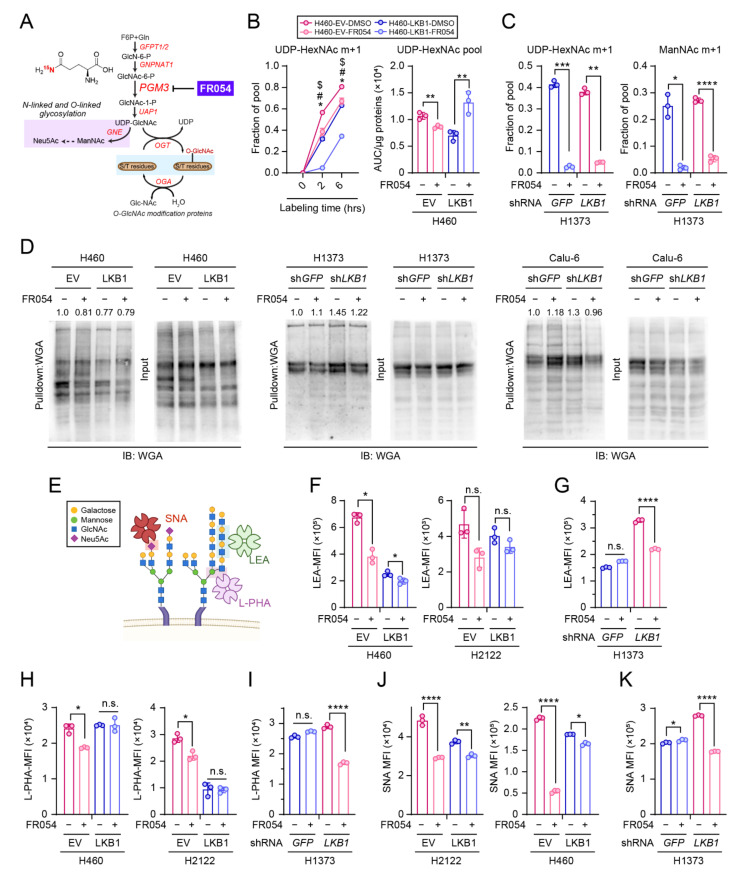
PGM3 inhibitor FR054 reduces the HBP flow in both KL and K cells but selectively suppresses glyco-functionalization pathways in KL co-mutant NSCLC cells. (**A**) Schematic of the HBP, including FR054, the inhibitor of PGM3. [γ-^15^N]glutamine is depicted. (**B**) **Left**: time course of ^15^N labelling in UDP-HexNAc in EV- and LKB1-expressing H460 cells cultured with [γ-^15^N]glutamine, treated with either DMSO (a vehicle control) or FR054 (50 µM, 3 days). **Right**: an abundance of UDP-HexNAc from labeling assay was measured by summing mass isotopologues, followed by protein normalization. (**C**) **Left**: ^15^N labeling in UDP-HexNAc was measured in an isogenic pair of H1373 cells treated with either DMSO or FR054 (100 µM, 3 days). **Right**: ^15^N labeling in ManNAc was measured in an isogenic pair of H1373 cells treated with either DMSO or FR054 (100 µM, 3 days). Cells were cultured with [γ-^15^N] glutamine for 6 h. (**D**) Effect of FR054 treatment on protein O-GlcNAcylation. One KL isogenic pair and two K isogenic pairs were used. WGA pulldown was performed (left/each cell line), and total protein extract before the addition of WGA was used as the input control (right/each cell line). Band intensity was quantified with Photoshop, and relative band intensity was obtained by calculating a ratio between each WGA pulldown band and input control. (**E**) Schematic for lectins and glycan structures. (**F**,**G**) Cell-surface LEA lectin binding was measured by flow cytometry in three isogenic pair cell lines with or without FR054 treatment for 3 days. (**H**,**I**) Cell-surface L-PHA lectin binding was measured by flow cytometry in three isogenic pair cell lines with or without FR054 treatment for 3 days. (**J**,**K**) Cell-surface SNA lectin binding was measured by flow cytometry in three isogenic pair cell lines with or without FR054 treatment for 3 days. (**B**) (left panel) Statistical significance was assessed using two-way ANOVA, followed by Tukey’s multiple comparisons test. * *p* < 0.05 compared to EV-FR054; ^#^ *p* < 0.05 compared to LKB1-DMSO; ^$^ *p* < 0.05 compared to LKB1-FR054. (**B**) (right panel) Statistical significance was assessed using two-tailed Student’s *t*-test/each isogenic pair. ** *p* < 0.01. (**C**,**F**–**K**) Statistical significance was assessed using two-tailed Student’s *t*-test/each isogenic pair. n.s., not significant; * *p* < 0.05; ** *p* < 0.01; *** *p* < 0.001; **** *p* < 0.0001. Targeted metabolomics and isotope tracing experiments were performed once. FACS analyses were performed twice.

**Figure 5 cells-11-00176-f005:**
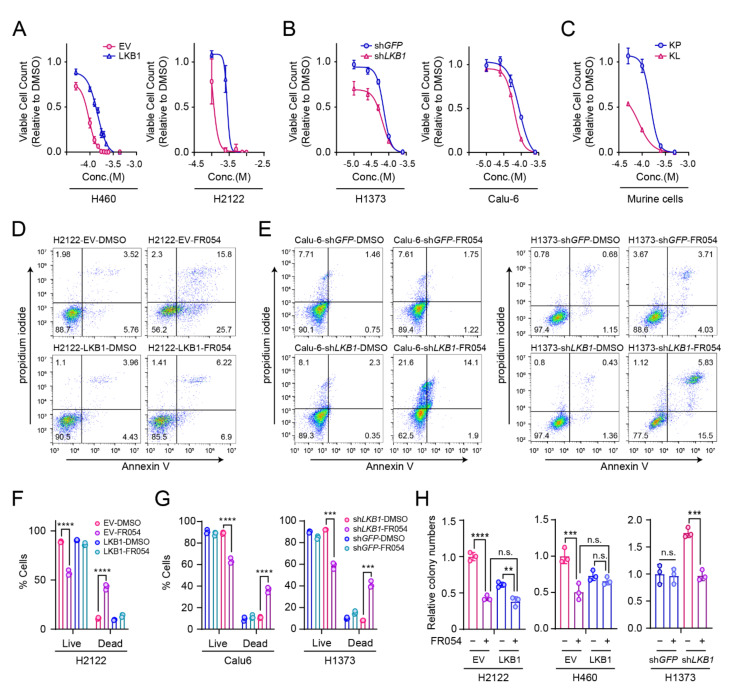
FR054 inhibits viability and clonogenicity of KL co-mutant NSCLC cells. (**A**–**C**) Sensitivity to *PGM3* silencing in K mutant and KL co-mutant cells. Two isogenic pairs of KL co-mutant cells (**A**), two isogenic pairs of K mutant cells (**B**), and murine NSCLC cells with either *KRAS/LKB1* co-mutations (KL) or *KRAS/TP53* co-mutations (KP) (**C**) were used. (**D**,**E**) Effect of FR054 treatment on cell death in K mutant cells and KL co-mutant cells. Three isogenic pairs were treated with either DMSO or FR054 (H460, 50 µM; H2122, 100 µM; H1373, 100 µM) for 3 days. Representative dot plots of FACS results/cell lines are shown. (**F**,**G**) Quantified FACS data from (**D**,**E**)**.** (**H**) Effect of *PGM3* knockout on anchorage-independent growth of K mutant cells and KL co-mutant cells (*n* = 3). FR054 concentration: H460, 50 µM; H2122, 100 µM; H1373, 100 µM. (**F**–**H**) Statistical significance was assessed using one-way ANOVA, followed by Tukey’s multiple comparisons test. n.s., not significant; ** *p* < 0.01; *** *p* < 0.001; **** *p* < 0.0001. Cell viability assays (*n* = 6), FACS analysis, and soft agar assays were performed twice.

**Figure 6 cells-11-00176-f006:**
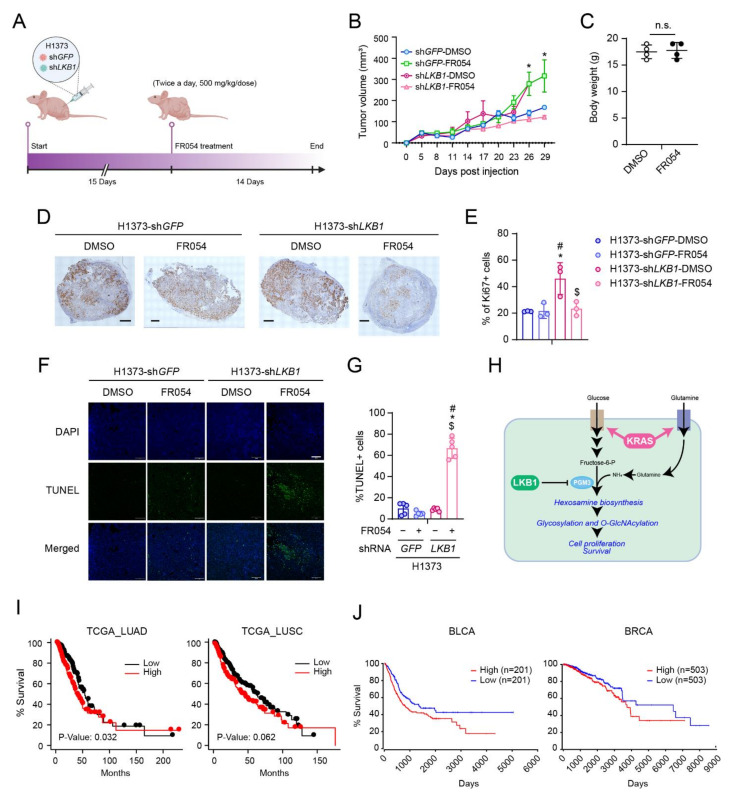
PGM3 inhibition reduces KL co-mutant NSCLC tumor growth in vivo. (**A**) Schematic diagram of the experimental procedure. sh*GFP*-expressing H1373 cells (1 × 10^6^cells) were injected into the left flank of the mouse, and sh*LKB1*-expressing H1373 cells (1 × 10^6^ cells) were injected into the right flank of the mouse. (**B**) Growth of sh*GFP*- and sh*LKB1*-expressing H1373 xenografts in the presence and absence of FR054 (500 mg/kg/dose, twice a day, 14 days total). Mean tumor volume and s.d. are shown for each group (*n* = 4). (**C**) Mouse weight with either vehicle control or FR054 at the day of euthanasia. (**D**) Representative Ki67 staining images of vehicle-treated and FR054-treated mice (*n* = 3/condition). Scale bar, 500 µm. (**E**) Ki67+ cells and total cells/tumor were quantified using Matlab. (**F**) Representative TUNEL staining of tumor tissues. 4′,6-diamidino-2-phenylindole (DAPI) was used to stain DNA. Scale bars, 100 μm. (**G**) TUNEL+ cells and total cells/tumor were quantified using Matlab. (**H**) Working model. Metabolic alterations mediated by concurrent mutations of KRAS and LKB1 created PGM3 dependence. (**I**) Kaplan-Meier plot associating *PGM3* mRNA expression with NSCLC (LUAD, lung adenocarcinoma (**left**) and LUSC, lung squamous carcinoma (**right**)) patient survival. Dataset is from Lung Cancer Explore, generated by UTSW (https://lce.biohpc.swmed.edu/lungcancer/, accessed on 21 November 2021). (**J**) Kaplan-Meier plot associating *PGM3* mRNA expression with bladder cancer (BLCA) and breast cancer (BRCA) patient survival. Dataset is from OncoLnc (http://www.oncolnc.org/, accessed on 21 November 2021). (**B**) Statistical significance was assessed using a two-way ANOVA with Tukey’s multiple comparisons test. * *p* < 0.05 compared to DMSO. (**C**) Statistical significance was assessed using paired Student’s *t*-test. n.s., not significant. (**E**,**G**) Statistical significance was assessed using one-way ANOVA with Tukey’s multiple comparisons test. * *p* < 0.05, compared to sh*GFP* with vehicle treatment, ^#^
*p* < 0.05, compared to sh*GFP* with FR054 treatment, ^$^
*p* < 0.05, compared to sh*LKB1* with vehicle treatment. Tumor growth experiment was performed once.

## Data Availability

Not applicable.

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
