# Peer review of "Targeting PGM3 as a Novel Therapeutic Strategy in KRAS/LKB1 Co-Mutant Lung Cancer"

_cells, 2022, doi:10.3390/cells11010176_

Round 1

Reviewer 1 Report

Lee et al. continued their previous work on KL mutant NSCLC tumor dependency on HBP. After showing KL mutant tumor cells depend on GFPT2, the authors discovered these cancer cells are also dependent on PGM3, which acts downstream of GFPT2. Although this is interesting finding, some questions need to be addressed:

  1. Does replenishing GlcN-6-P rescue the phenotype of GFPT2 knockdown or azaserine treatment in KL mutant cells?
  2. What about GlcNAc-1-P for PGM3 loss or FR054 treatment?
  3. What about GNPNAT1? Do KL cells show dependency on GNPNAT1 as well?
  4. It seems in H460, re-expressing LKB1did not fully rescue the dependency of glycosylation on PGM3, which is different from GFPT2. Any thoughts on that?
  5. Any known off-target effects of FR054? 50uM is very high concentration.
  6. Fig 6B color scheme is not clear, can't tell shGFP-DMSO from shGFP-FR054. But anyway in H1373 LKB1 WT cells, FR054 also had growth inhibition effect? similar to when LKB1 is knocked down suggesting significant off-target effects of FR054. What about KD PGM3 in vivo? Will it be only effective in shLKB1 cells? 

Reviewer 2 Report

In this study, the authors examined a specific metabolic pathway in KRas/LKB1 co-mutant NSCLC. In addition, they demonstrated the efficacy of PGM3-targeted therapy in lung cancer. These findings are based on their previous study, which demonstrated that the PGM3 inhibitor, FR054, effectively suppressed KRas/LKB1 co-mutant NSCLC both in vitro and in vivo, whereas the GFPT2 inhibitor showed a limited anti-tumor effect.

The manuscript is well written. However, the issues raised in comments below need to be addressed to improve the manuscript.

The comments are as follows:

  1. In this study, five human lung cancer cell lines, A549, H460, Calu-6, H2122, and H1373, having different KRas mutations, were used for in vitro A549 cells harbor KRasG12S, H460 cells harbor KRasQ61H, Calu-6 cells harbor KRasQ61K, whereas both H2122 and H1373 cells harbor KRasG12C.
  • The authors should describe the KRAS mutation status of these types of NSCLC cells in the Materials and Methods section.
  • The authors should also state whether the difference in mutation status affects the activity of the hexosamine biosynthesis pathway (HBP) with or without LKB1 mutation.
  1. As explained in the manuscript, H1313 cells are the LKB1 wild-type, while H2122 cells are the LKB1-mutant NSCLC cells.
  • Was there an observed difference in their sensitivity to the FDA-approved KRAS inhibitors, Sotorasib and Adagrasib?
  • Moreover, did FR054 show a more effective decrease in H2112 cellular viability in vitro than the KRasG12C inhibitors?

In order to prove the superiority of FR054, it might be beneficial to carry out those experiments.

  1. Could FR054 induce apoptosis in H1373-shLKB1 cells in vivo (Figure 6)?

Round 2

Reviewer 1 Report

The revised version and the authors' response has addressed the concerns.